# Comparison of lean mass indices as predictors of mortality in incident peritoneal dialysis patients

**Seok Hui Kang, A. Young Kim, Jun Young Do**[ID]*

Division of Nephrology, Department of Internal Medicine, Yeungnam University Medical Center, Daegu, Republic of Korea

* jydo@med.yu.ac.kr

## Abstract

### Background

Few studies have considered optimal adjusted lean mass indices for prediction of clinical outcomes in peritoneal dialysis (PD) patients. We aimed to evaluate clinical variables using various adjusted indices in PD patients.

### Methods

Total 528 incident PD patients were included. Lean mass was measured using dual energy X-ray absorptiometry. Appendicular lean mass (ALM) was calculated using the sum for both upper and lower extremities. Each ALM index was calculated using ALM per body weight (ALM/BW), height squared (ALM/Ht$^2$), or body mass index (ALM/BMI). Limb/trunk lean mass (LTLM) ratio was defined as the sum for both upper and lower extremities divided by trunk lean mass.

### Results

A total of 528 patients were analyzed men: 286, women: 242. In area under the receiver operating characteristic curve analyses, LTLM alone was associated with 1 year mortality. In the LTLM ratio, the cut-off value for 1-year mortality was ≤ 0.829 in men and ≤ 0.717 in women, respectively. In both sexes, LTLM ratio alone showed statistical significance in all-cause mortality in both univariate and multivariate Cox-regression analyses. Compared with other indices, the LTLM ratio was independent of edema and fat in both sexes. Edema- and C-reactive protein-adjusted correlation analysis showed that LTLM ratio alone was associated with serum albumin in men. Although statistical significance was not obtained for women, the correlation coefficient was highest for the LTLM ratio compared with other indices.

### Conclusion

Among various indices using lean mass, LTLM ratio was independent of volume status and fat mass and was associated with mortality in incident PD patients.

**Data Availability Statement:** All data generated or analyzed during this study are included in this published article and its Supplementary Information files.

**Funding:** This work was supported by the Medical Research Center Program (2015R1A5A2009124) through the National Research Foundation of Korea funded by the Ministry of Science, ICT, and Future Planning. The funder had no role in the study design; the collection, analysis, and interpretation of data; the writing of the report; and the decision to submit the article for publication.

**Competing interests:** The authors have declared that no competing interests exist.

## Introduction

End-stage renal disease patients require renal replacement therapy, and peritoneal dialysis (PD) is an important option for long-term maintenance dialysis. Registries have shown that among incident dialysis patients, PD accounts for 9.6% in the USA, 16.0% in Europe, and 7.5% in Korea [1–3]. Many factors related to uremia (e.g., inflammation, anorexia, and hypercatabolism) and PD per se (e.g., loss of nutrients into dialysate, peritonitis, and bio-incompatible dialysate) are involved in development of protein energy wasting, which lead to decreased muscle mass and/or sarcopenia [4]. The prevalence of sarcopenia is high, compared with that in the general population [5]. The development of sarcopenia is associated with adverse outcomes in PD patients [5]. Therefore, criteria for the diagnosis of sarcopenia in PD patients are needed to improve prognosis.

Dual energy X-ray absorptiometry (DXA) is a reliable tool used to predict muscle mass. Appendicular lean mass (ALM) by DXA can be adjusted for body size. Various methods have adjusted for body weight, height squared, or body mass index in predicting muscle mass [6–9]. However, other factors than those applicable to general or elderly populations must be considered in predicting low muscle mass using DXA in PD patients. PD patients are over-hydrated compared with the general population, lead to overestimation of muscle mass and underestimation of sarcopenia [10]. However, muscle mass indices have only been validated in a population with stable volume. The limb/trunk lean mass (LTLM) ratio was first reported by Kato et al., as a nutritional and prognostic indicator in hemodialysis patients [11]. Lean mass can be associated with overhydration, when measured using DXA, but the ratio using lean limb mass and trunk lean mass would attenuate the effect of overhydration. In addition, malnourished individuals, such as PD patients, have increased protein catabolism earlier in the extremities than in the viscera [12]. These findings may reveal that the LTLM ratio can be an option for predicting clinical outcomes in patients with PD rather than classic lean mass indices. Few studies have considered optimal adjusted indices for the prediction of clinical outcomes in PD patients. We aimed to evaluate clinical variables using various adjusted indices in PD patients.

## Subjects and methods

### Study population

Our study was a retrospective observational study, conducted at Yeungnam University medical center in Korea between January 2001 and March 2016. DXA for lean mass measurement is routinely performed in our center, at 1 month following PD initiation, after obtaining informed consent. Of 694 incident PD patients, 166 who did not have baseline DXA measurements or sufficient laboratory data were excluded.

### Ethics statement

The institutional review board of Yeungnam University medical center approved our study (No. 2021-01-019). All personal identifiers were deleted prior to analysis. Therefore, the board waived the need for informed consent. The study was conducted in accordance with the principles that have their origin in the Declaration of Helsinki.

### Study variables

We collected the following data 4 weeks after PD initiation: age, sex, body weight, height, comorbidities, body mass index, residual renal function (RRF, mL·min$^{-1}$·1.73 m$^{-2}$), C-reactive protein level (mg/dL), serum albumin level (g/dL), modality (continuous ambulatory PD or

automated PD), peritoneal membrane characteristics, weekly Kt/Vurea, visceral fat area ($cm^3$), the edema index, lean mass, and fat mass.

Comorbidities were classified using the Davies risk index, and graded as low, intermediate, or high risk [13]. Body mass index was calculated using body weight by height squared ($kg/m^2$). The RRF was calculated based on 24-hour urine collection as follows: RRF = $\frac{24hr\ urine\ creatinine\ (mg/dL)}{serum\ creatinine\ (mg/dL)} + \frac{24hr\ urine\ urea\ nitrogen\ (mg/dL)}{serum\ urea\ nitrogen\ (mg/dL)} \times 0.5 \times$ (urine volume/1440) $\times$ 1.73/body surface area ($m^2$) [14]. A modified equilibration test was performed to determine peritoneal membrane characteristics. Dialysate containing 4.25% glucose was infused and drained after 4 hours. Dialysate/plasma creatinine ratio was calculated using the following formula = $\frac{4hr\ dialysate\ creatinine\ (mg/dL)}{Plasma\ creatinine\ (mg/dL)}$.

A ratio of > 0.81 was defined as a high transporter status. Weekly Kt/Vurea was calculated based on 24-hour urine and dialysate as follows: Weekly Kt/Vurea = 7 $\times$ $\frac{[24hr\ urine\ nitrogen\ (mg/dL) \times 24hr\ urine\ volume\ (L)] + [24hr\ dialysate\ urea\ nitrogen\ \left(\frac{mg}{dL}\right) \times 24hr\ drain\ volume\ (L)]}{Distribution\ volume\ of\ urea\ (L) \times serum\ urea\ nitrogen\ (mg/dL)}$. Watson's formula was used to estimate the distribution volume of urea [15]. An automatic chemical analyzer (AU4500; Olympus, Tokyo, Japan) was used to estimate serum albumin and C-reactive protein levels. The bromocresol green method was used to estimate serum albumin level.

Lean mass and fat mass were measured using DXA. For the DXA assessments, dialysate was drained from the abdomen prior to measurement. Body composition was measured using DXA with the subject supine and clothed with a light gown. The images were obtained from a Discovery QDR Series bone densitometer (Hologic, Madison, WI, USA). The scans were analyzed using the Hologic Discovery Wi software (version 13.3). Calibration of the densitometer was checked daily using a manufacturer supplied standard calibration block and passed at –1.5 ~ +1.5% control limits. All regions of interest were measured by a technician according to the manufacturer's manual [16]. Briefly, the upper extremities were defined superiorly by the horizontal shoulder line, medially by the vertical arm line bisecting the glenoid fossa, and laterally by the border of the global regions of interest. Low extremities were defined by the oblique femoral line superomedially, by the vertical leg line inferomedially and laterally by the vertical line on the lateral aspect of the leg. The rib region was defined superiorly by the shoulder line, laterally by the vertical arm lines, medially by the vertical spine lines, and inferiorly by the line at the iliac crest. The pelvis was defined superiorly by the horizontal line at the iliac crest and laterally and inferiorly by the oblique lines passing through the center of the femoral neck. The trunk was defined as the sum of measures in both the ribs and pelvis. Within observer measurement was performed for the intraclass correlations of the appendicular lean, trunk lean, and total fat masses. Intraclass correlation coefficients between the 2 measurements of the appendicular lean, trunk lean, and total fat masses were 0.999 (95% confidence interval [CI], 0.997–1.000; $P < 0.001$), 0.998 (95% CI, 0.996–0.999; $P < 0.001$), and 0.999 (95% CI, 0.998–1.000; $P < 0.001$), respectively. As the estimates in our study were measured by one technician, measurements between the observers were not identified.

Visceral fat area and edema index were measured using bioimpedance analysis. The edema index was calculated as extracellular fluid per total body fluid and measured using the Inbody Body Composition Analyzer version 4.0 (Biospace, Seoul, Korea), with the subject in standing position. The Inbody Body Composition Analyzer 4.0 is a multi-frequency bioimpedance analysis using impedance at 1, 5, 250, 500, and 1000 kHz [17]. This measures total body fluid and extracellular fluid. Previous studies validated estimates of total body fluid and extracellular fluid using Inbody Body Composition Analyzer [18,19]. Only few data are available regarding the accuracy and precision of Inbody estimates using standard methods such as the dilution

method in PD patients. However, we compared measurements between Inbody and bioimpedance spectrometry as relatively validated methods in 41 PD patients. The results showed that correlation coefficients between bioimpedance spectrometry and Inbody measurements were 0.944 for extracellular volume, 0.907 for intracellular volume, and 0.872 for edema index ($P < 0.001$ for all variables).

## Definitions of muscle mass indices

ALM was calculated using the sum for both upper and lower extremities. Each ALM index was calculated using ALM per body weight (ALM/BW), height squared (ALM/Ht$^2$), or body mass index (ALM/BMI). The LTLM ratio was defined as the sum for both upper and lower extremities divided by trunk lean mass.

Cut off values for low muscle mass were defined as the lowest quintile of each index for a Korean young adult population. We analyzed data from the Korea National Health and Nutrition Examination Survey 2009–2011. This database is a nationwide, multi-stage, stratified survey of a representative sample of the South Korean population conducted by the Korea Centers for Disease Control and Prevention. The numbers of total participants was 37,753. Among these, participants <20 years old or >39 years old, with chronic disease (diabetes mellitus, chronic kidney disease, hypertension, stroke, coronary artery disease, thyroid disease, arthritis, pulmonary tuberculosis, asthma, liver cirrhosis, or any malignancies) and without DXA data were excluded. Finally, 1,861 men and 2,656 women were analyzed to define cut-off values. For men, cut-off values for ALM, ALM/Ht$^2$, ALM/BW, ALM/BMI, and LTLM ratio were 21.0, 7.11, 30.7, 0.903, and 0.773, respectively. For women, values for ALM, ALM/Ht$^2$, ALM/BW, ALM/BMI, and LTLM ratio were 12.9, 5.09, 24.2, 0.607, and 0.650, respectively. All laboratory tests, anthropometry, DXA, and bioimpedance analysis, included in this study, were performed on the same day.

## Outcome measures

All mortality events were retrieved from patient medical records. Patients with kidney transplantation, transfer to hemodialysis, recovery of renal function, or transfer to other hospitals were defined as censored data at the end of PD.

## Statistical analyses

SPSS version 23 (Chicago, IL, USA) software was used to analyze the data. Categorical data and continuous data were expressed as counts (percentages) and mean ± standard deviation, respectively. Categorical data and means were analyzed using Pearson's chi-square test and Student's *t*-test, respectively. The area under the receiver operating characteristic curve (AUROC) was used to calculate the probability of predict death at 1 year after PD initiation, cutoff values, sensitivity, and specificity. The best cutoff value was calculated using the Youden index in the AUROC. MedCalc version 11.6.1.0 software (MedCalc, Mariakerke, Belgium) was used for AUROC.

Correlations were analyzed to assess the strength of the relationships between continuous variables. We performed Cox regression analyses for survival. For multivariate analyses, we adjusted for age, the Davies risk index, weekly Kt/Vurea, RRF, C-reactive protein, and the edema index. Multivariate Cox regression analyses were performed using the enter method. For these, censored cases were defined as survivors at the end of follow-up. The proportional hazard assumption was satisfied for all the variables. For competing risk analyses, we defined censored cases as competing risk and performed the Fine and Gray competing risk model,

using SAS version 9.4 (SAS Institute, Cary, NC, USA). A $P$-value < 0.05 was considered statistically significant.

## Results

### Baseline characteristics of participants

A total of 528 patients were analyzed of whom 286 were men and 242 were women (Table 1). More men than women had a high Davies risk index and underwent automated PD; RRF value was also greater in men. Body mass index, C-reactive protein level, serum albumin level, edema index, and follow-up duration were similar between the sexes. Although 23.9% of the participants were excluded from our study, there were no significant differences between the included and excluded participants (S1 Table).

### AUROC analyses of various indices

Participants with data for survival or death in the year after PD initiation were included in the AUROC analysis (Fig 1). In men, AUROC values for ALM, ALM/Ht$^2$, ALM/BW, ALM/BMI, and LTLM were 0.570 (95% CI, 0.508–0.629; $P$ = 0.267), 0.576 (95% CI, 0.515–0.636; $P$ = 0.254), 0.566 (95% CI, 0.504–0.625; $P$ = 0.381), 0,582 (95% CI, 0.521–0.641; $P$ = 0.241), and 0.675 (95% CI, 0.616–0.731; $P$ = 0.003), respectively (Fig 1A). In women, corresponding values were 0.595 (95% CI, 0.528–0.659; $P$ = 0.175), 0.552 (95% CI, 0.485–0.618; $P$ = 0.509), 0.504 (95% CI, 0.437–0.570; $P$ = 0.956), 0.546 (95% CI, 0.479–0.612; $P$ = 0.495), and 0.654 (95% CI, 0.589–0.716; $P$ = 0.030), respectively (Fig 1B). LTLM alone was associated with 1 year mortality. In the LTLM ratio, the cut-off value for 1-year mortality was ≤ 0.829 in men and ≤ 0.717 in women. At this cut-off point, sensitivity and specificity were 85% and 49% in men and 72.2% and 58.3% in women, respectively. At the cut-off point for the Korean young adult population, sensitivity and specificity were 25% and 86.9% in men and 22.2% and 95.3% in women, respectively.

**Table 1. Clinical characteristics of participants at the time of peritoneal dialysis initiation.**

|  | Men (n = 286) | Women (n = 242) | REF | Patients outside the REF (men, %) | Patients outside the REF (women, %) | $P$-value* |
|---|---|---|---|---|---|---|
| Age (years) | 53.9 ± 13.1 | 53.1 ± 14.0 | – |  |  | 0.496 |
| Body mass index (kg/m$^2$) | 23.7 ± 3.0 | 23.4 ± 3.7 | 18.5–24.9 | 94 (32.9%) | 90 (37.2%) | 0.389 |
| RRF (mL·min$^{-1}$·1.73 m$^{-2}$) | 4.7 ± 3.9 | 3.2 ± 2.6 | – |  |  | <0.001 |
| Serum albumin (g/dL, normal) | 3.48 ± 0.59 | 3.49 ± 0.51 | 3.5–5.0 | 131 (45.8%) | 110 (45.5%) | 0.920 |
| C-reactive protein (mg/dL) | 0.66 ± 1.37 | 0.74 ± 2.02 | 0–0.5 | 71 (24.8%) | 45 (18.6%) | 0.620 |
| Modality (APD) | 62 (21.7%) | 25 (10.3%) | – |  |  | <0.001 |
| Edema index | 0.368 ± 0.036 | 0.367 ± 0.023 | 0.30–0.35 | 214 (74.8%) | 182 (75.2%) | 0.770 |
| High transporter | 35 (12.2%) | 31 (12.8%) | – |  |  | 0.843 |
| Weekly Kt/Vurea | 2.24 ± 0.77 | 2.58 ± 0.66 | – |  |  | <0.001 |
| Follow-up duration (mon) | 52.7 ± 40.2 | 57.8 ± 41.9 | – |  |  | 0.151 |
| Davies risk index |  |  | – |  |  | 0.045 |
| Low | 86 (30.1%) | 93 (38.4%) |  |  |  |  |
| Intermediate | 179 (62.6%) | 140 (57.9%) |  |  |  |  |
| High | 21 (7.3%) | 9 (3.7%) |  |  |  |  |

Data are expressed as numbers (percentages) for categorical variables and as median ± standard deviation for continuous variables.

*$P$-values were tested using the Student's $t$-test for continuous variables and Pearson's $\chi^2$ or Fisher's exact test for categorical variables.

Abbreviations: RRF, residual renal function; APD, automated peritoneal dialysis; REF, reference range.

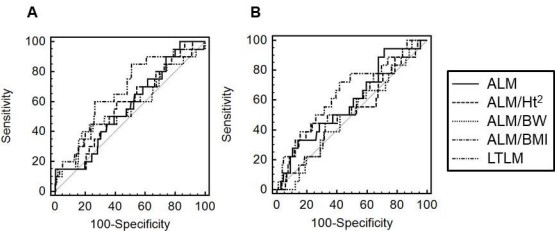

**Fig 1.** AUROC values for various indices used in prediction of mortality at 1 year after PD initiation (A: men, B: women). Abbreviations: AUROC, area under the receiver operating characteristic curve; PD, peritoneal dialysis; ALM, appendicular lean mass; ALM/Ht$^2$, appendicular lean mass per height squared; ALM/BW, appendicular lean mass per body weight; ALM/BMI, appendicular lean mass per body mass index; LTLM, limb/trunk lean mass ratio.

## Association between various indices and clinical findings

Edema index was positively correlated with ALM/BW and ALM/BMI in men and all indices except LTLM ratio in women (Table 2). Total fat mass and visceral fat area were positively correlated with ALM and ALM/Ht$^2$ in both sexes, and inversely correlated with ALM/BW in both sexes. ALM/BMI was inversely correlated with total fat mass in both sexes. Serum albumin was positively correlated with LTLM ratio in men and inversely correlated with ALM/Ht$^2$ and ALM/BMI in women.

Partial correlation analysis with adjustment for the edema index and C-reactive protein was performed to identify associations between various indices and serum albumin. Correlation coefficients for ALM, ALM/Ht$^2$, ALM/BW, ALM/BMI, and LTLM were 0.113 ($P = 0.062$), 0.068 ($P = 0.261$), −0.027 ($P = 0.655$), 0.054 ($P = 0.372$), and 0.172 ($P = 0.004$) in men and 0.032 ($P = 0.632$), −0.006 ($P = 0.928$), −0.032 ($P = 0.633$), 0.015 ($P = 0.817$), and 0.090 ($P = 0.175$) in women, respectively.

## Survival analyses

The number of participants who survived, died, or were censored at end point of follow-up were 152, 209, and 167, respectively. Among the censored patients, the cause of censoring was

**Table 2. Correlation between various indices and clinical variables in PD patients.**

| | Edema index | | Total fat mass | | Visceral fat area | | Serum albumin | |
|---|---|---|---|---|---|---|---|---|
| | r | P-value* | r | P-value* | r | P-value* | r | P-value* |
| Men | | | | | | | | |
| ALM | 0.081 | 0.170 | 0.250 | <0.001 | 0.470 | <0.001 | 0.085 | 0.157 |
| ALM/Ht$^2$ | 0.105 | 0.075 | 0.204 | 0.001 | 0.473 | <0.001 | 0.038 | 0.521 |
| ALM/BW | 0.202 | 0.001 | −0.391 | <0.001 | −0.175 | 0.033 | −0.099 | 0.097 |
| ALM/BMI | 0.146 | 0.013 | −0.216 | <0.001 | −0.031 | 0.710 | −0.008 | 0.894 |
| LTLM ratio | −0.086 | 0.148 | 0.034 | 0.567 | −0.111 | 0.179 | 0.211 | <0.001 |
| Women | | | | | | | | |
| ALM | 0.253 | <0.001 | 0.228 | 0.000 | 0.437 | <0.001 | −0.096 | 0.139 |
| ALM/Ht$^2$ | 0.263 | <0.001 | 0.187 | 0.004 | 0.467 | <0.001 | −0.139 | 0.032 |
| ALM/BW | 0.289 | <0.001 | −0.495 | <0.001 | −0.214 | 0.018 | −0.157 | 0.016 |
| ALM/BMI | 0.274 | <0.001 | −0.366 | <0.001 | −0.152 | 0.096 | −0.106 | 0.105 |
| LTLM ratio | 0.031 | 0.634 | −0.059 | 0.362 | −0.013 | 0.890 | 0.046 | 0.478 |

*P-values were tested using Pearson's correlation.

Abbreviations: PD, peritoneal dialysis; r, correlation coefficient; ALM, appendicular lean mass; ALM/Ht$^2$, appendicular lean mass per height squared; ALM/BW, appendicular lean mass per body weight; ALM/BMI, appendicular lean mass per body mass index; LTLM, limb/trunk lean mass ratio.

as follows: 91 patients were transferred for hemodialysis (54.5%), 51 for kidney transplantation (30.5%), 22 were transferred to other hospitals (13.2%), and 3 had recovery of their renal function (1.8%).

The hazard ratio (95% CI) for each increase of 1 unit for each index is shown in Table 3. In men, LTLM ratio alone showed statistical significance in all-cause mortality in both univariate and multivariate Cox-regression analyses. In women, ALM/BW, ALM/BMI, and LTLM ratio showed statistical significance in both univariate and multivariate Cox-regression analyses.

When we used cut-off values for each Korean young adult population index, the prevalence of low lean mass based on ALM, ALM/Ht$^2$, ALM/BW, ALM/BMI, and LTLM was 62.6%, 54.9%, 55.2%, 62.9%, and 34.3% in men and 31.0%, 14.0%, 33.9%, 47.9%, and 19.0% in women, respectively. Cox regression analyses using these categorical data also showed a similar trend (Table 4). For competing risk analyses, the hazard ratio (95% CI) for one unit increase of each index is shown in S2 Table. Results from the competing risk analysis were similar to those from Cox regression analyses performed using the total cohort.

## Discussion

We analyzed data according to sex. Death at 1 year after PD initiation was associated with LTLM alone in both sexes. Compared with other indices, the LTLM ratio was independent of edema and fat in both sexes. Edema- and C-reactive protein-adjusted correlation analysis showed that LTLM ratio alone was associated with serum albumin in men. Although statistical significance was not obtained for women, the correlation coefficient was highest for the LTLM ratio compared with other indices. Survival analyses using continuous or categorical variables showed that LTLM ratio alone was associated with mortality in both sexes.

Compared with the general population, PD patients show some differences in lean mass measurements using DXA. First, PD patients are over-hydrated compared with the general population or hemodialysis patients [20]. A previous study showed that lean mass measurement using DXA is volume dependent [10]. Over-hydration in PD patients is associated with over-estimation of lean mass. Second, PD patients undergo glucose loading by dialysate and

**Table 3. Univariate and multivariate hazard ratios for all-cause mortality according to various indices.**

| Independent variables | Univariate | | Multivariate | |
|---|---|---|---|---|
| | Hazard ratio (95% CI) | *P*-value | Hazard ratio (95% CI) | *P*-value* |
| Men (per increase 1 unit) | | | | |
| ALM | 0.903 (0.852–0.956) | <0.001 | 0.967 (0.904–1.034) | 0.327 |
| ALM/Ht$^2$ | 0.800 (0.667–0.958) | 0.015 | 0.857 (0.696–1.055) | 0.146 |
| ALM/BW | 0.945 (0.892–1.002) | 0.057 | 0.966 (0.906–1.029) | 0.286 |
| ALM/BMI | 0.056 (0.011–0.290) | 0.001 | 0.613 (0.090–4.168) | 0.617 |
| LTLM ratio | 0.009 (0.001–0.060) | <0.001 | 0.062 (0.005–0.733) | 0.027 |
| Women (per increase 1 unit) | | | | |
| ALM | 0.922 (0.855–0.994) | 0.035 | 0.952 (0.874–1.037) | 0.263 |
| ALM/Ht$^2$ | 0.866 (0.707–1.062) | 0.168 | 0.820 (0.648–1.037) | 0.097 |
| ALM/BW | 0.914 (0.866–0.965) | 0.001 | 0.911 (0.853–0.973) | 0.005 |
| ALM/BMI | 0.017 (0.002–0.131) | <0.001 | 0.059 (0.005–0.742) | 0.028 |
| LTLM ratio | 0.006 (0.001–0.048) | <0.001 | 0.018 (0.002–0.207) | 0.001 |

*Multivariable analysis was adjusted for age, the Davies risk index, weekly Kt/Vurea, residual renal function, C-reactive protein, and edema index.

Abbreviations: CI, confidence interval; ALM, appendicular lean mass; ALM/Ht$^2$, appendicular lean mass per height squared; ALM/BW, appendicular lean mass per body weight; ALM/BMI, appendicular lean mass per body mass index; LTLM, limb/trunk lean mass ratio.

**Table 4. Univariate and multivariate hazard ratio for all-cause mortality according to the low group for each lean mass index.**

| Independent variables | Univariate | | Multivariate | |
|---|---|---|---|---|
| | Hazard ratio (95% CI) | *P*-value | Hazard ratio (95% CI) | *P*-value* |
| Men (ref: high group) | | | | |
| ALM | 1.676 (1.122–2.503) | 0.012 | 1.129 (0.714–1.784) | 0.603 |
| ALM/Ht$^2$ | 1.232 (0.850–1.785) | 0.270 | 1.118 (0.734–1.703) | 0.604 |
| ALM/BW | 1.562 (1.071–2.279) | 0.020 | 1.060 (0.691–1.626) | 0.790 |
| ALM/BMI | 1.919 (1.280–2.877) | 0.002 | 0.960 (0.604–1.526) | 0.863 |
| LTLM ratio | 2.397 (1.651–3.480) | <0.001 | 1.655 (1.105–2.481) | 0.015 |
| Women (ref: high group) | | | | |
| ALM | 1.682 (1.091–2.593) | 0.019 | 1.598 (0.996–2.564) | 0.052 |
| ALM/Ht$^2$ | 1.718 (1.012–2.915) | 0.045 | 1.895 (1.030–3.487) | 0.040 |
| ALM/BW | 1.653 (1.092–2.503) | 0.017 | 1.822 (1.165–2.849) | 0.009 |
| ALM/BMI | 2.563 (1.663–3.949) | <0.001 | 2.503 (1.504–4.163) | <0.001 |
| LTLM ratio | 2.475 (1.542–3.973) | <0.001 | 1.854 (1.090–3.153) | 0.023 |

*Multivariable analysis was adjusted for age, the Davies risk index, weekly Kt/Vurea, residual renal function, C-reactive protein, and edema index.

Abbreviations: CI, confidence interval; ALM, appendicular lean mass; ALM/Ht$^2$, appendicular lean mass per height squared; ALM/BW, appendicular lean mass per body weight; ALM/BMI, appendicular lean mass per body mass index; LTLM, limb/trunk lean mass ratio.

develop insulin resistance under uremic conditions, leading to fat accumulation [21]. Third, both muscle mass and fat mass are considered nutritional markers in PD patients. Therefore, optimal indicators for ALM should be independent of fat mass, obesity, and volume status.

Body size should be considered in evaluation of the absolute amount of ALM. ALM can be adjusted using various body size indicators such as height squared, body weight, or body mass index [22]. However, these are correlated with fat mass or obesity [22]. ALM/Ht$^2$ is positively associated with obesity, and can be overestimated in patients with high fat mass or obesity. Our data also showed a positive association between ALM/Ht$^2$ and total fat mass or visceral fat area. ALM/BW and ALM/BMI indicators are adjusted for obesity. However, these have an inverse association with obesity, leading to loss of ALM as a prognostic factor. Our data also showed an inverse association between ALM/BW or ALM/BMI and total fat mass. Consequently, ALM indices adjusted for anthropometric indicators would be inaccurate for estimation of absolute ALM in PD patients with high fat mass. These indicators are also positively associated with the edema index, leading to overestimation of lean mass according to volume status, especially in women.

Recent studies have investigated the clinical impact of body composition measurements. Previous studies using large hemodialysis patient cohorts showed that lean tissue and/or fat tissue indices are associated with favorable survival [23,24]. A meta-analysis using 6 studies, which investigated the association between clinical outcome and variables from bioimpedance spectroscopy in hemodialysis patients, showed that volume overload and a low lean tissue index were associated with high mortality [25]. Some studies demonstrated a positive association between body composition measurements and clinical outcomes in PD patients [26,27]. However, there are no definitive guidelines for an optimal index predicting clinical outcome in volume dependent-dialysis patients. Previous guidelines have recommended cut-off values using various adjustment methods such as height, weight, or body mass index [6,8,9]. However, these values were obtained from the general population or non-volume dependent populations and these adjustments may not be optimal in PD patients with a high-fat mass and volume overload. Our study showed that the LTLM ratio did not correlate with the edema

index in either sex and was independent of fat mass or visceral fat area, because anthropometric indicators were not used for adjustment. In addition, we evaluated the association between all the indices and serum albumin level as a nutritional marker. As serum albumin is inversely associated with inflammation and volume status, we performed partial correlation analysis with adjustment for C-reactive protein and edema index [28,29]. LTLM was positively correlated with serum albumin level in men. For women, the correlation coefficient was highest for LTLM, but without statistical significance. Finally, survival analyses were performed for each index. Cox regression analyses revealed that LTLM ratio was associated with mortality in PD patients (both sexes). Our study illustrates that the LTLM ratio may be the most optimal for predicting the nutritional status and prognosis among various adjustment variables in PD patients.

We aimed to determine the predictive value of each variable for mortality in patients with PD. We performed ROC analyses using death at one year after PD initiation. Although the AUROC was small, the LTLM ratio alone was associated with death at one year after PD initiation in both sexes. The cut-off value for the LTLM ratio using AUROC analyses in PD patients was greater than that in the lowest quintile of the Korean young adult population. Sensitivities using ≤ 0.829 in men and ≤ 0.717 in women were higher than those using the lowest quintile in the Korean young adult population. Malnutrition or low muscle mass is more common in PD patients compared with general population. Their presence is associated with high mortality in PD patients, and screening is important to improve prognosis. Cut-off values with high sensitivity may be more useful. Further investigation is needed to identify accurate cut-off values for prediction of poor prognosis in PD patients.

Differences in ethnicity should be considered for defining cut-off values of the LTLM ratio. In our study, the mean values of the LTLM ratio in a young Korean population were 0.817 ± 0.003 in men and 0.694 ± 0.003 in women. Cut-off values for low muscle mass, defined as below 20th percentile of the LTLM ratio, were 0.773 in men and 0.650 in women. The cut-off value for 1-year mortality was 0.829 in men and 0.717 in women. As both Korean and Japanese individuals are of Asian ethnicity, we suggest that cut-off values for the LTLM ratio may be similar between the two ethnicities. However, there are limited data regarding the cut-off value of low lean mass for the LTLM ratio using a young Japanese population with a large sample size. A Japanese study on prevalent hemodialysis patients showed that the mean values of the LTLM ratio were 0.692 in men and 0.644 in women [11]. However, their study was on patients who underwent hemodialysis for 1–27 years and our study enrolled incident PD patients. This discrepancy may be associated with the difference in dialysis vintage and modality. PD patients are more volume overloaded than hemodialysis patients, and long dialysis duration would lead to decreased muscle mass compared with incident dialysis. Further data on young Japanese adults would be required to clarify whether data from our study are suitable for a Japanese population.

Measurements using bioimpedance spectroscopy are a little different from those using bioimpedance analysis in the present study. Fresenius's body composition monitor using multi-frequency bioimpedance spectroscopy is used with a broadband of frequencies, such as 5 kHz to 1000 kHz. This measures total body water and extracellular water. However, the Inbody 4.0 measures extracellular fluid and total body fluid. The definitions of fluid and water were ambiguous, but definitely had notable differences [30]. Fluid consists of pure water and solutes, such as electrolytes and protein, dissolved within water [31–33]. Intracellular fluid contains more solutes than extracellular fluid. Therefore, fluid is positively associated with water, but extracellular water/total body water is slightly greater than extracellular fluid/total body fluid. Data from a normal population showed that the normal values of extracellular water/total body water and extracellular fluid/total body fluid were approximately 0.38 and

0.33, respectively [31,34,35]. The manufacturer suggests that the participants with ≥ 0.35 of the edema indices from extracellular fluid/total body fluid would have edema, but there were few data regarding definitive cut-off values for predicting edema. Regardless of the limited data for the cut-off value, previous studies demonstrated a positive association between the edema index and clinical outcomes [26,36,37]. Further investigations are required to identify significant cut-off values for predicting edema.

This study had several limitations. First, our study design used a retrospective cohort and was single-center based. In addition, our data did not include information regarding muscle strength or physical performance. Such data are important indicators for prediction of optimal muscle mass measurement. Second, 23.9% of the participants were excluded from our analysis because of insufficient data. This may be associated with a selection bias. Therefore, we compared the clinical characteristics at baseline, between the included and excluded participants and there were no significant differences between the two groups. This would be helpful to attenuate the selection bias by excluding 23.9% of the participants. Third, serum albumin levels were measured using the bromocresol green method. Serum albumin measurements using bromocresol green are associated with non-specific binding and a proportional bias between the two colorimetric and immunologic methods has been reported [38]. A previous study showed that the bromocresol green method led to a mean bias of 0.62 g/dL compared to the immunologic method, which resulted in an inappropriate treatment decision for 59% of patients with hypoalbuminemia [39]. Although bromocresol green is a classical method for predicting serum albumin level, the use of bromocresol purple or immunologic method should be preferred to measure the serum albumin level to avoid overestimation in patient at a risk of malnutrition. A prospective, multicenter study including additional parameters such as hand-grip strength, gait speed, or a short physical performance battery, and more precise laboratory results, such as serum albumin level using bromocresol purple or immunologic method, is needed to identify a more clear association and overcome limitations.

In conclusion, among various indices using lean mass, LTLM ratio was independent of volume status and fat mass and was associated with mortality in incident PD patients. These findings imply that screening or monitoring using LTLM ratio may be necessary to predict the prognosis in PD patients.

## Supporting information

**S1 Table. Comparison of clinical characteristics between included and excluded participants at the time of peritoneal dialysis initiation.**
(DOCX)

**S2 Table. Univariate and multivariate hazard ratios using competing risk model according to various indices.**
(DOCX)

## Acknowledgments

A part of this study was presented in a poster presentation at the 38st Annual Meeting of the Korean Society of Nephrology and 55[th] European Renal Association-European Dialysis Transplant Association. The abstract for the poster was included in the conference proceeding which was peer-reviewed. Abstract details available at: http://www.ksn.or.kr/file/journal/701525348/2018/PDL%20047.pdf or https://academic.oup.com/ndt/article/33/suppl_1/i201/4997589.

## Author Contributions

**Conceptualization:** Seok Hui Kang.

**Data curation:** Seok Hui Kang, A. Young Kim.

**Formal analysis:** Seok Hui Kang.

**Funding acquisition:** Jun Young Do.

**Investigation:** Seok Hui Kang.

**Methodology:** Seok Hui Kang, A. Young Kim.

**Software:** Jun Young Do.

**Supervision:** Jun Young Do.

**Validation:** Jun Young Do.

**Writing – original draft:** Seok Hui Kang.

**Writing – review & editing:** Jun Young Do.

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
