## [Decision Letter · Decision Letter 0]

4 May 2021

PONE-D-21-09966

Comparison of lean mass indices as predictors of mortality in incident peritoneal dialysis patients

PLOS ONE

Dear Dr. Young Do

Thank you for submitting your manuscript to PLOS ONE. After careful consideration, we feel that it has merit but does not fully meet PLOS ONE’s publication criteria as it currently stands. Therefore, we invite you to submit a revised version of the manuscript that addresses the points raised during the review process.

We look forward to receiving your revised manuscript.

Kind regards,

Pasqual Barretti, Ph.D., MD

Academic Editor

PLOS ONE

Journal Requirements:

2. We note that the abstract of your article may have been presented elsewhere. At this time, please clarify whether your abstract was previously published, and whether it was peer-reviewed. If previously published, please provide a link to the full text of the abstract or a PDF copy and clarify the copyright permissions on the published work.

3. Thank you for providing the date(s) when patient medical information was initially recorded.

Please also include the date(s) on which your research team accessed the databases/records to obtain the retrospective data used in your study.

4. Please provide the name of the institution where participants were from.

Additional Editor Comments:

Both authors have raised important questions, which must be addressed by the authors. According to my point of view, those questions are easy to address, as well as the suggestions were made. The reviewer 1 have emphasized statistical and methodological aspects, in special on censored patients and competitive risks to outcomes. The reviewer 2 have made important questions about nutritional aspects.

Reviewers' comments:

Reviewer's Responses to Questions

**Comments to the Author**

1. Is the manuscript technically sound, and do the data support the conclusions?

Reviewer #1: Yes

Reviewer #2: Yes

2. Has the statistical analysis been performed appropriately and rigorously? 

Reviewer #1: No

Reviewer #2: Yes

3. Have the authors made all data underlying the findings in their manuscript fully available?

Reviewer #1: Yes

Reviewer #2: Yes

4. Is the manuscript presented in an intelligible fashion and written in standard English?

Reviewer #1: Yes

Reviewer #2: Yes

5. Review Comments to the Author

Reviewer #1: Dear editor and authors, I carefully reviewed the manuscript entitled “Comparison of lean mass indices as predictors of mortality in incident peritoneal dialysis patients”. My comments and concerns are the following:

The research question is interesting, and the manuscript well written in a standard technical English.

The rationale of the study is well elaborated but I suggest to change the information that DEXA is commonly used because the low access of most centers to this technology. Instead, I’d use DEXA is a reliable tool indicated to … or is commonly used in science to predict …

Almost 25% of the study population were excluded because they didn’t have a DEXA measurement. It is important to characterize these individuals in comparison to the population included to facilitate the analysis of any potential selection bias.

The authors should include in their limitation the albumin measured with bromocresol green instead of purple.

Censoring in dialysis studies is key for interpretation of survival analysis. As performed in current studies there is a risk that competing events may have influenced the final results and interpretation. So, two recommendations: first make available the absolute number of patients who were censored and the reasons for censoring; and perform a competing risk analysis considering as censored only patients active at the end of follow-up and the remaining as a competing event.

The study design (retrospective) should be described also in the methods and not only in the paragraph of limitations.

Reviewer #2: Review comments to the author

The authors performed a retrospective observational analysis to determine clinical variables (using various adjusted indices) as predictors of mortality in incident peritoneal dialysis. There are some recommendations that might be useful to consider:

a) In the introduction, the authors discuss in a more emphasized way about the ALM, which is not the main result of this study.

The main result of this study is not highlighted. It's important to talk about LTLM.

b) In the methods, the authors need to clarify the study design.

c) About evaluations:

- Were all evaluations performed at the same time or was there a time between them? (DXA, BIS, echocardiography, anthropometry and laboratory evaluation).

d) The text contains many abbreviations. eliminate some in order to make reading more comfortable.

e) the discussion answers many questions about the study, however, it seems confusing at times. Review is needed.

6. PLOS authors have the option to publish the peer review history of their article (what does this mean?). If published, this will include your full peer review and any attached files.

Reviewer #1: No

Reviewer #2: No

---

## [Author Response · Author response to Decision Letter 0]

28 May 2021

Additional Editor Comments:

Both authors have raised important questions, which must be addressed by the authors. According to my point of view, those questions are easy to address, as well as the suggestions were made. The reviewer 1 have emphasized statistical and methodological aspects, in special on censored patients and competitive risks to outcomes. The reviewer 2 have made important questions about nutritional aspects.

Answer: Thank you for your comments regarding our manuscript. We have incorporated the reviewers’ suggestions into the revised manuscript and indicated them in red font. We have responded to the reviewers’ comments. We hope that all issues have been adequately addressed and that the manuscript is now acceptable for publication.

Reviewer #1: Dear editor and authors, I carefully reviewed the manuscript entitled “Comparison of lean mass indices as predictors of mortality in incident peritoneal dialysis patients”. My comments and concerns are the following:

The research question is interesting, and the manuscript well written in a standard technical English.

The rationale of the study is well elaborated but I suggest to change the information that DEXA is commonly used because the low access of most centers to this technology. Instead, I’d use DEXA is a reliable tool indicated to … or is commonly used in science to predict …

Answer: Thank you for your comments. We have revised the relevant line as follows: Dual X-ray absorptiometry (DXA) is a reliable tool used to predict the muscle mass.

Almost 25% of the study population were excluded because they didn’t have a DEXA measurement. It is important to characterize these individuals in comparison to the population included to facilitate the analysis of any potential selection bias.

Answer: We agree with the reviewer’s comments. Exclusion of 23.9% of the participants could be associated with a selection bias. We compared the two cohorts (included vs. excluded participants), but there were no significant differences in the baseline characteristics. These results can be helpful in attenuating the selection bias by the exclusion of participants. We have added these comments in the Limitation section and Table S1 for comparison between the included and excluded participants.

The authors should include in their limitation the albumin measured with bromocresol green instead of purple.

Answer: As the reviewer pointed out, serum albumin measurements using bromocresol green is associated with non-specific binding and a proportional bias among the two colorimetric and immunologic methods has been reported [1]. A previous study showed that the bromocresol green method led to a mean bias of 0.62 g/dL compared to the immunologic method, which resulted in an inappropriate treatment decision for 59% of patients with hypoalbuminemia [2]. Although bromocresol green is a classical method for predicting serum albumin level, the use of bromocresol purple or immunologic method should be preferred to measure the serum albumin level, to avoid overestimation in patients with risk of malnutrition. We have added these comments and references in the Limitation section.

Added reference

[1] Alcorta MD, Alvarez PC, Cabetas RN, Martín MA, Valero M, Candela CG. The importance of serum albumin determination method to classify patients based on nutritional status. Clin Nutr ESPEN. 2018 Jun;25:110-113.

[2] van de Logt AE, Rijpma SR, Vink CH, Prudon-Rosmulder E, Wetzels JF, van Berkel M. The bias between different albumin assays may affect clinical decision-making. Kidney Int. 2019 Jun;95(6):1514-1517.

Censoring in dialysis studies is key for interpretation of survival analysis. As performed in current studies there is a risk that competing events may have influenced the final results and interpretation. So, two recommendations: first make available the absolute number of patients who were censored and the reasons for censoring; and perform a competing risk analysis considering as censored only patients active at the end of follow-up and the remaining as a competing event.

Answer: Thank you for your comment. In our study, the number of participants who survived, died, or were censored at the end-point of the follow-up were 152, 209, and 167, respectively. Among the censored patients, the cause of censoring was as follows: 91 patients were transferred for hemodialysis (54.5%); 51 for kidney transplantation (30.5%); 22 were transferred to other hospitals (13.2%); and 3 had recovery of their renal function (1.8%). We defined censored cases as competing risk and performed Fine and Gray competing risk model with cohorts without censoring.

Results from competing risk model were similar with those from the total cohort. We have added these comments and Table S2 in the Methods and Results sections.

The study design (retrospective) should be described also in the methods and not only in the paragraph of limitations.

Answer: Thank you for your comment. We have indicated “retrospective observational study” in the Methods section.

Reviewer #2: Review comments to the author

The authors performed a retrospective observational analysis to determine clinical variables (using various adjusted indices) as predictors of mortality in incident peritoneal dialysis. There are some recommendations that might be useful to consider:

a) In the introduction, the authors discuss in a more emphasized way about the ALM, which is not the main result of this study. The main result of this study is not highlighted. It's important to talk about LTLM.

Answer: Thank you for your comments. We have added some comments regarding the LTLM ratio in the Introduction section and transferred some comments from the Discussion section to Introduction.

The following comments have been added: The limb/trunk lean mass (LTLM) ratio was first reported by Kato et al. as a nutritional and prognostic indicator in hemodialysis patients [1]. Lean mass can be associated with over-hydration when measured using DXA, but the ratio using lean limb mass and trunk lean mass would attenuate the effect of over-hydration. In addition, malnourished individuals, such as PD patients, have increased protein catabolism in the extremities earlier than in the viscera [2]. These may reveal that LTLM ratio can be an option for predicting clinical outcomes in PD patients compared to classic lean mass indices.

Reference

[1] Kato A, Odamaki M, Yamamoto T, Yonemura K, Maruyama Y, Kumagai H, et al. Influence of body composition on 5 year mortality in patients on regular haemodialysis. Nephrol Dial Transplant 2003;18:333-340.

[2] Heymsfield SB, McManus C, Stevens V, Smith J. Muscle mass: reliable indicator of protein-energy malnutrition severity and outcome. Am J Clin Nutr. 1982 May;35(5 Suppl):1192-1199.

b) In the methods, the authors need to clarify the study design.

Answer: Thank you for your comments. We have indicated “retrospective observational study” in the Methods section.

c) About evaluations:

- Were all evaluations performed at the same time or was there a time between them? (DXA, BIS, echocardiography, anthropometry and laboratory evaluation).

Answer: Thank you for your comment. All laboratory studies, anthropometry, DXA, and bioimpedance analysis measurements, included in this study, were performed on the same day. We have added the comment in the Method section.

d) The text contains many abbreviations. eliminate some in order to make reading more comfortable.

Answer: Thank you for your comment. We have replaced some abbreviations with their full name throughout the manuscript.

e) the discussion answers many questions about the study, however, it seems confusing at times. Review is needed.

Answer: Thank you for your comment. We have condensed the Discussion section. Part of the discussion which were repeated or were similar have been merged. We have also deleted redundant or unnecessary portions, such as difference between bioimpedance spectroscopy and bioimpedance analysis.

---

## [Decision Letter · Decision Letter 1]

7 Jul 2021

Comparison of lean mass indices as predictors of mortality in incident peritoneal dialysis patients

PONE-D-21-09966R1

Dear Dr. Jung Young Do

We’re pleased to inform you that your manuscript has been judged scientifically suitable for publication and will be formally accepted for publication once it meets all outstanding technical requirements.

Kind regards,

Pasqual Barretti, Ph.D., MD

Academic Editor

PLOS ONE

Additional Editor Comments (optional):

The paper has consistently improved and the authors have addressed all the questions. I agree with the reviewer's decision.

Reviewers' comments:

Reviewer's Responses to Questions

**Comments to the Author**

1. If the authors have adequately addressed your comments raised in a previous round of review and you feel that this manuscript is now acceptable for publication, you may indicate that here to bypass the “Comments to the Author” section, enter your conflict of interest statement in the “Confidential to Editor” section, and submit your "Accept" recommendation.

Reviewer #1: All comments have been addressed

Reviewer #2: All comments have been addressed

2. Is the manuscript technically sound, and do the data support the conclusions?

Reviewer #1: Yes

Reviewer #2: Yes

3. Has the statistical analysis been performed appropriately and rigorously? 

Reviewer #1: Yes

Reviewer #2: Yes

4. Have the authors made all data underlying the findings in their manuscript fully available?

Reviewer #1: Yes

Reviewer #2: Yes

5. Is the manuscript presented in an intelligible fashion and written in standard English?

Reviewer #1: Yes

Reviewer #2: Yes

6. Review Comments to the Author

Reviewer #1: Dear editor/authors,

All my comments, critics and suggestions were properly addressed improving the final quality of the manuscript.

I have no further comments

Reviewer #2: The authors have adequately addressed all comments raised in the review and it is acceptable for publication.

7. PLOS authors have the option to publish the peer review history of their article (what does this mean?). If published, this will include your full peer review and any attached files.

Reviewer #1: **Yes: **Thyago Proenca de Moraes

Reviewer #2: No

---

## [Editor Report · Acceptance letter]

13 Jul 2021

PONE-D-21-09966R1 

Comparison of lean mass indices as predictors of mortality in incident peritoneal dialysis patients 

Dear Dr. Do:

I'm pleased to inform you that your manuscript has been deemed suitable for publication in PLOS ONE. Congratulations! Your manuscript is now with our production department. 

Kind regards, 

on behalf of

Prof. Pasqual Barretti 

Academic Editor

PLOS ONE